**Data Availability Statement:** All relevant data are within the paper.

# Isolation, identification, and antimicrobial susceptibility pattern of *Campylobacter jejuni* and *Campylobacter coli* from cattle, goat, and chicken meats in Mekelle, Ethiopia

Yohans Hagos[1], Getachew Gugsa🔘[2]*, Nesibu Awol[2], Meselu Ahmed[2], Yisehak Tsegaye🔘[3], Nigus Abebe[3], Abrha Bsrat[3]

1 Shire Agricultural Technical Vocational and Education Training College, Shire, Tigray, Ethiopia,
2 Department of Veterinary Medicine, School of Veterinary Medicine, Wollo University, Dessie, Ethiopia,
3 Department of Veterinary Medicine, College of Veterinary Sciences, Mekelle University, Mekelle, Ethiopia

◎ These authors contributed equally to this work.
* gugsag@yahoo.com

## Abstract

*Campylobacter jejuni* and *Campylobacter coli* are globally recognized as a major cause of bacterial foodborne gastroenteritis. A cross-sectional study was conducted from October 2015 to May 2016 in Mekelle city to isolate, identify, and estimate the prevalence of *C. jejuni* and *C. coli* in raw meat samples and to determine their antibiotic susceptibility pattern. A total of 384 raw meat samples were randomly collected from bovine (n = 210), goat (n = 108), and chicken (n = 66), and isolation and identification of *Campylobacter spp.* were performed using standard bacteriological techniques and PCR. Antibiotic susceptibility test was performed using disc diffusion method. Of the total 384 raw meat samples, 64 (16.67%) were found positive for *Campylobacter spp.* The highest prevalence of *Campylobacter spp.* was found in chicken meat (43.93%) followed by bovine meat (11.90%) and goat meat (9.25%). The most prevalent *Campylobacter spp.* isolated from meat samples was *C. jejuni* (81.25%). The overall prevalence of *Campylobacter* in restaurants, butcher shops, and abattoir was 43.93%, 18.30%, and 9.30%, respectively. 96.8%, 81.25%, 75%, and 71% of the *Campylobacter spp.* isolates were sensitive to norfloxacin, erythromycin, chloramphenicol, and sulphamethoxazole-trimethoprim, respectively. However, 96.9%, 85.9%, and 50% of the isolates were resistant to ampicillin, amoxicillin, and streptomycin, respectively. Strains that developed multi-drug resistant were 68.7%. The result of this study revealed the occurrence of *Campylobacter* in bovine, goat, and chicken meats. Hence, there is a chance of acquiring infection via consumption of raw or undercooked meat. Thus, implementation of hygienic practices from a slaughterhouse to the retailers, proper handling and cooking of foods of meat are very important in preventing *Campylobacter* infection.

**Funding:** This research work was funded by College of Veterinary Sciences, Mekelle University.

**Competing interests:** The authors have declared that no competing interests exist.

## Introduction

Foodborne diseases occur as a result of the consumption of contaminated foodstuffs especially from animal products such as meat from infected animals or carcasses contaminated with pathogenic bacteria [1, 2]. Food-producing animals are the major reservoirs for many foodborne pathogens such as *Campylobacter species*, non-Typhi serotypes of *Salmonella enterica*, Shiga toxin-producing strains of *Escherichia coli*, and *Listeria monocytogenes*. Foodborne pathogens cause millions of cases of sporadic illness and chronic complications, as well as large and challenging outbreaks in many countries and between countries [3].

Worldwide, pathogenic *Campylobacter species* are the leading cause of bacterial-derived foodborne disease and are responsible for the cause of over 400–500 million infections cases each year [4–6]. *Campylobacter species* are normally carried in the intestinal tracts of many domestic livestock such as poultry, cattle, sheep, goat, pigs, as well as wild animals and birds [7–10]. Fecal matter is a major source of contamination and could reach carcasses through direct deposition [11]. Animal food products can become contaminated by this pathogen during slaughtering and carcass dressing [12]. Humans are infected by ingestion of undercooked or decontaminated meat, or handling of raw products or cross-contamination of raw to cooked foods, swimming in natural waters, direct contact with contaminated animals or animal carcasses, and traveling [13–15].

Pathogenic *Campylobacter spp*. known to be implicated in human infections include *C. jejuni*, *C. concisus*, *C. rectus*, *C. hyointestinalis*, *C. insulaenigrae*, *C. sputorum*, *C. helveticus*, *C. lari*, *C. fetus*, *C. mucosalis*, *C. coli*, *C. upsaliensis*, and *C. ureolyticus* [6]. Of these,*C.jejuni*and *C. coli* are considered the most commonly reported zoonosis in humans and recognized as the most common causative agents of bacterial gastroenteritis in the world [16–19].

Moreover, *Campylobacter* with resistance to antimicrobial agents has also been implicated worldwide [4, 20, 21]. The use of antimicrobial agents in food animals has resulted in the emergence and dissemination of antimicrobial-resistant bacteria including antimicrobial-resistant *Campylobacter*, which has a potentially serious impact on food safety in both animal and human health. The situation seems to deteriorate more rapidly in developing countries where there is widespread and uncontrolled use of antibiotics [22].

In Ethiopia, few studies were conducted on the prevalence and antimicrobial susceptibility of enteric Campylobacteriosis of human beings [23–25] and food of animal origins [9, 20, 26–28]. The absence of a national surveillance program, limited routine culture availability for the isolation of *Campylobacter spp*. in clinical and research settings, and the need for selective media and unique growth atmosphere make it difficult to give an accurate picture of the burden of the disease in Ethiopia. This fact indicates that *Campylobacter* as a zoonosis is not given appropriate weight and consideration, particularly in the current study area. As a result, the objectives of this study were to isolate and identify *C. jejuni* and *C. coli* from the meat of cattle, goat, and chicken collected from an abattoir, butcher shops, and restaurants in Mekelle City, estimate their prevalence and determine the antibiotic susceptibility pattern of *C.jejuni*and *C. coli* isolates.

## Material and methods

### Ethics approval

This study was reviewed and approved by the Research Ethics Committee of the College of Veterinary Sciences, Mekelle University.

### Study area

The study was conducted from October 2015 to May 2016 at an abattoir, butcher shops, and restaurants of Mekelle City. Mekelle is the capital city of Tigray National Regional State of

Ethiopia where thousands of cattle and goats are accessible from different districts of the region and the neighboring regions of the country for slaughter. Mekelle is found at 39°29' East and 13°30' North of the equator which is 783 kilometers away from Addis Ababa, which is the capital city of Ethiopia. The altitude of the area ranges from 2000–2200 meters above sea level. The mean annual rainfall of the area is 628.8 mm and an annual average temperature of 21°C. The city has seven sub-cities and a total population of 215,546 [29], 308 cafeterias, 292 restaurants, 258 supermarkets, and an active urban-rural exchange of goods which has 30000 micro-and small enterprises [30].

## Study design

A cross-sectional study was employed from October 2015 to May 2016 to isolate, identify, and estimate the prevalence and antibiotic susceptibility patterns of *C.jejuni* and *C. coli* from bovine, goat, and chicken meat samples collected from the abattoir, butcher shops, and restaurants.

## Sample size and sample collection

A total of 384 raw ready-to-eat meat samples comprising of cattle (n = 210), goat (n = 108),and chicken (n = 66) meats were collected from the abattoir (n = 258), butcher shops (n = 60), and restaurants (n = 66) of the study area. All samples were placed in polyethylene plastic bags to prevent spilling and cross-contamination and immediately transported to the Molecular Biology Laboratory of the College of Veterinary Sciences, Mekelle University using an icebox with ice packs.

## Bacteriological isolation and identification of *Campylobacter species*

Approximately 10 grams of raw meat sample was aseptically collected using sterile forceps and scissor and placed into 90ml of buffered peptone water in a sterile plastic bag and homogenized for 1 minute using a stomacher (Lab Blender 400, Seward Medical, London, England) and incubated at 37°C for 48h in the microaerophilic atmosphere (gas mix of 5% $O_2$, 10% $CO_2$, and 85% $N_2$). Then a 0.1ml of the enriched sample was streaked onto Karmali Campylobacter Agar Base (HiMedia Laboratories, Mumbai, India)(Blood free *Campylobacter* selective agar base medium containing *Campylobacter* selective supplement comprising cefoperazone, amphotericin B (CCDA selective supplement SR0155E)) [31] and kept in a gas jar containing *Campylobacter* gas pack systems to maintain the microaerophilic condition and was incubated at a temperature of 37°C for 48h. The colonies were provisionally identified based on staining reaction with Gram's stain, cellular morphology [32], catalase test, and oxidase test [33], and growth appearance on 5% sheep blood agar (Oxoid Ltd., Basingstoke, Hampshire, England) at 37°C after 24 h [34].

All the thermophilic *Campylobacters* isolates were tested for Hippurate hydrolysis, $H_2S$ production, and susceptibility to Nalidixic acid and Cephalothin as proposed by [34]. Susceptibility tests to Nalidixic acid (30 μg) and Cephalothin (30 μg) were performed using the standard agar disc diffusion method as recommended by Clinical and Laboratory Standards Institutions (CLSI) and isolates were categorized as sensitive and/or resistant according to the interpretation table of the [35]. The presumed *Campylobacter* isolates were preserved in brain heart infusion broth supplemented with15% glycerol in Eppendorf tubes at -20°C for further analysis. Bacterial strains that were used as quality control organisms in this study were standard strains of *S. aureus*, *S. agalactiae*, and *E. coli* obtained from the National Veterinary Institute (NVI), Debre-Zeit.

## Polymerase chain reaction for detection of *mapA* and *ceuE* genes

The genomic DNA of the pheno typically resembled isolates of *Campylobacter* was extracted using the Phenol Chloroform method (Phenol: Chloroform: Isoamyl alcohol mixture (24:25:1)) according to [36]. Then 20μl of each extracted genomic DNA sample was run in an agarose gel electrophoresis and visualized under UV-light gel doc. Then after, a genome-based polymerase chain reaction (PCR) was done as described by [37] using the following species-specific primers: F-5'CTA TTT TAT TTT TGA GTG CTT GTG3' and R-5'GCT TTA TTT GCC ATTT GTT TTA TTA3' was used to amplify the *mapA* gene of *C. jejuni*(589 bp), and F-5'ATT TGA AAA TTG CTC CAA CTA TG3' and R-5'TGA TTT TAT TAT TTG TAG CAG CG3' were used to amplify the *ceuE* gene of *C. coli*(462 bp). Each PCR reaction mixture was performed in a 50μl total volume containing 10μl of template DNA,5μl of 5X PC buffer, 5μl of $MgCl_2$, 1μl of each of the primers, 0.75μl of 10 mM of each dNTPs, 0.15μl of Taq DNA polymerase, and27.1μl nuclease-free distilled water. Amplification was carried out with thermal cycling conditions of an initial denaturation at 95˚C for 5 min followed by 45 cycles of denaturation at 94˚C for 35s, annealing at 54˚C for 35s and extension at 72˚C for 35s, and with a final extension at 72˚C for 6 min. Finally, the PCR products were separated by running on a 1.5% (w/v) agarose gel containing 0.3mg/ml ethidium bromide. Electrophoresis was conducted in a horizontal equipment system for 120 min at 90 V using 1X TAE buffer (40 mM Tris, 1 mM EDTA, and 20 mM glacial acetic acid, pH 8.0). The amplicons were visualized under UV-light gel doc and their molecular weights were estimated by comparing with100bp DNA molecular weight marker(Solis BioDyne, Tartu, Estonia).

## Antimicrobial susceptibility testing

The *Campylobacter spp.* isolates were screened for in vitro antimicrobial susceptibility using the standard agar disc diffusion method as recommended by Clinical and Laboratory Standards Institutions (CLSI) on Mueller-Hinton agar supplemented with 5% sheep blood (Oxoid Ltd., Basingstoke, Hampshire, England). The following nine different antibiotic discs, with their concentrations given in parentheses, were used in the antibiogram testing: Amoxicillin (AML)(10μg), Ampicillin (AMP)(10μg), Chloramphenicol (C)(30μg), Erythromycin (E) (15μg), Gentamycin (CN)(10μg), Norfloxacin (NOR)(10μg), Streptomycin (S)(10μg), Tetracycline (TE)(30μg), and Sulfamethoxazole-trimethoprim (SXT)(25μg) (Oxoid Company, Hampshire, England). After 48h of microaerophilic incubation at 37˚C, the clear zones (inhibition zones of bacterial growth around the antibiotic discs(including the discs)diameter for individual antimicrobial agents were measured and then translated into Sensitive (S), Intermediate (I), and Resistant (R) categories according to the interpretation table of the Clinical and Laboratory Standard Institute [35].

## Data storage and statistical analysis

All collected data were entered into Microsoft Excel Sheet (Microsoft Corp., Redmond, WA, USA) and analyzed using SPSS version 20 statistical computer software program. Chi-square ($\chi^2$) test and Logistic regression were applied to assess the associations. For all tests, a *p-value* of less than 0.05 was considered statistically significant.

## Results

### The overall prevalence of *Campylobacter* species

Out of the total of 384 collected meat samples, 64(16.67%) were positive for the two *Campylobacter spp.* The highest (43.93%) and lowest (9.25%) prevalence of *Campylobacter spp.* were

**Table 1. Prevalence of *Campylobacter spp.* among different sample types and sources.**

| Risk Factors | No. of Examined | No. of Positive (%) | $\chi^2$ | P-value |
|---|---|---|---|---|
| **Type of meat** | | | | |
| Goat meat | 108 | 10(9.25) | 43.04 | 0.000 |
| Cattle meat | 210 | 25(11.90) | | |
| Chicken meat | 66 | 29(43.93) | | |
| **Total** | **384** | **64(16.67)** | | |
| **Sources of meat** | | | | |
| Abattoir | 258 | 24(9.30) | 45.53 | 0.000 |
| Butcher | 60 | 11(18.33) | | |
| Restaurants | 66 | 29(43.93) | | |
| **Total** | **384** | **64(16.67)** | | |

recorded in samples taken from chicken (found to be 7.68 times more likely to have *Campylobacter* contamination compared to other sample types) and goat, respectively. Whereas, the highest (43.93%) and lowest (9.25%) prevalence were recorded in meat samples collected from restaurants (found to be four times more likely to have *Campylobacter* contamination compared to other sample sources) and abattoir, respectively. Both sample types and sources had significant differences (p = 0.00; $\chi^2$ = 43.04or OR = 7.68, CI = 3.40–17.30,and p = 0.00; $\chi^2$ = 45.53 or OR = 7.64, CI = 4.01–14.52, respectively) in the prevalence of the two *Campylobacter spp.* as it is shown in Tables 1 and 2 below.

## Contamination rate of *C.jejuni* and *C. coli* in the different sample types

Of the two *Campylobacter spp.* isolated and identified from cattle, goat, and chicken meat samples *C. jejuni* and *C. coli* accounted for 81.25% and 18.75%, respectively. The prevalence of *C. jejuni* and *C.coli* in cattle, goat, and chicken meat samples were found to be 76% and 24%, 80% and 20%, and 86.21%and 13.79%, respectively (Table 3).

## PCR amplification results of *Campylobacter spp.* isolates

Besides the phenotypic characterization, PCR amplification of the 64 samples revealed that 52(81.25%) of the isolates were *C.jejuni*(having a molecular weight of 589 bp) and the remaining 12(18.75%) isolates were *C.coli* (having a molecular weight of462 bp) to the targeted genes.

**Table 2. Logistic regression analysis results of sample types and sample sources.**

| Types of sample | No of examined | No of positive (%) | OR(95%CI) | P-value |
|---|---|---|---|---|
| Goat meat* | 108 | 10(9.25) | 1 | |
| Cattle meat | 210 | 25(11.90) | 1.32(0.61–2.86) | 0.476 |
| Chicken meat | 66 | 29(43.93) | 7.68(3.40–17.30) | 0.000 |
| **Total** | **384** | **64(16.67)** | | |
| Abattoir* | 258 | 24(9.30) | 1 | |
| Butcher | 60 | 11(18.33) | 2.18(1.00–4.76) | 0.048 |
| Restaurants | 66 | 29(43.93) | 7.64(4.01–14.52) | 0.000 |
| **Total** | **384** | **64(16.67)** | | |

OR = Odd ratio; CI = Confidence interval.

**Table 3. The contamination rate of *C. jejuni* and *C. coli* among different sample types.**

| Sample Type Prevalence | *Campylobacter spp.* | |
|---|---|---|
| | *C.jejuni* | *C.coli* |
| Cattle meat (n = 25) | 19(76%) | 6(24%) |
| Goat meat (n = 10) | 8(80%) | 2(20%) |
| Chicken meat (n = 29) | 25(86.21%) | 4(13.79%) |
| Total (n = 64) | 52(81.25%) | 12(18.75%) |

## Antimicrobial susceptibility pattern of *Campylobacter spp.* isolates

*Campylobacter spp.* isolated from the different sample types and sources were susceptible to Norfloxacin (96.8%), Erythromycin (81.25%), Chloramphenicol (75%), and Gentamycin (75%). However, the isolates had shown resistance to Ampicillin (96.9%) and Amoxicillin (85.9%) (Table 4). Moreover, 96.8% of the isolates developed resistance for two or more than two drugs as it is shown in Fig 1.

Species based antibiogram result of the isolates revealed that the highest level of sensitivity of both *C. jejuni* (98.1%) and *C. coli* (91.7%)was observed against Norfloxacin. Whereas both *C.jejuni*(96.2%) and *C. coli*(100%) were showed the highest level of resistance against Ampicillin as it is shown in Table 5.

## Discussion

In the current study, the overall prevalence of *Campylobacter spp.* was found to be 16.67%. The highest prevalence was found in chicken meat samples (43.93%). Chicken meats were found to be 7.68 times more likely to have *Campylobacter* when compared to goat and cattle meat. The difference in the prevalence of *Campylobacter* between different types of meat samples was found to be statistically significant ($p < 0.05$) (OR = 7.68, CI = 3.40–17.30). The prevalence of *Campylobacter spp.* in chicken meat samples was 43.93% which was comparable with those reported by [38], 44%, [39], 56.1%,and [40], 48.02% in Iran. This was higher than the report of [41] who reported a prevalence of 1.93%. However, the present finding was lower than studies conducted by [42–44], and [45] who reported the prevalence of 61.7% and 70.7% and 65% and 81.3% of *Campylobacter spp.* in Ahvaz, Iran; Washington; Northern Ireland, and Northern Italy, respectively. It is a well-known fact that poultry appeared to be a significant source of *Campylobacter* and chicken was found to be heavily intestinal carriers of *Campylobacter* when compared with other food animals [8]. Wide variation (0–90%) in the prevalence of

**Table 4. In vitro antimicrobial sensitivity pattern of *Campylobacter spp.* isolates.**

| Type of antibiotics | Interpretations | | |
|---|---|---|---|
| | Susceptible (%) | Intermediate (%) | Resistant (%) |
| Ampicillin | 0(0) | 2(3.1) | 62(96.9) |
| Amoxicillin | 3(4.7) | 6(9.4) | 55(85.9) |
| Chloramphenicol | 48(75) | 5(7.8) | 11(17.2) |
| Erythromycin | 52(81.25) | 1(1.6) | 11(17.2) |
| Gentamycin | 48(75) | 8(12.5) | 8(12.5) |
| Norfloxacin | 62(96.8) | 1(1.6) | 1(1.6) |
| Streptomycin | 25(39.06) | 7(10.9) | 32(50) |
| Tetracycline | 42(65.6) | 6(9.4) | 16(25) |
| Sulfamethoxazole-Trimethoprim | 46(71.8) | 4(6.25) | 14(21.9) |

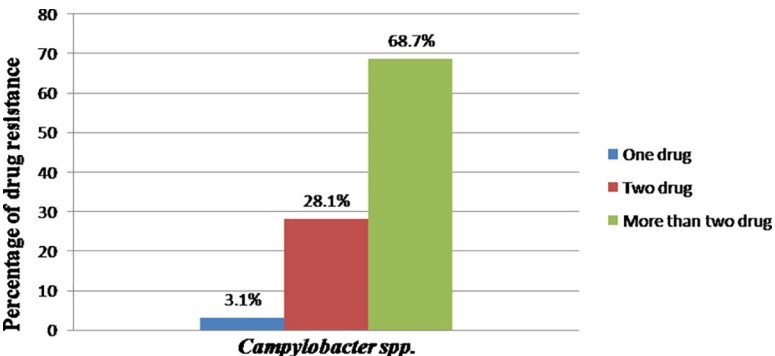

**Fig 1. Percentage of drug resistance of *Campylobacter spp*.**

*Campylobacter* in fresh poultry meat had been reported in different countries [46–48]. These variations in *Campylobacter spp*. prevalence might be due to differences in hygienic conditions, cross-contamination that may occur during de-feathering, eviscerating, and some other environmental factor such as the temperature of water in the scalding tank.

In this study, the prevalence of *Campylobacter spp*. in bovine meat was 11.90%. This was comparable to the finding reported from a previous study done by [10] (12.9%) in Nigeria, and [49] (10%) in Iran. However, it was higher than the findings reported by [9] (6.2%) in Ethiopia [50]; (5.6%) in Morogoro, Tanzania; and [51] (0.8%)in Australia. Food of animal origin has been incriminated for being the main source of *Campylobacter* infection in humans [52]. Since raw meat from beef is widely consumed in the country; the occurrence of *Campylobacter* in meat increases the likelihood of the pathogen transmission to humans. The present finding was lower than studies conducted by [10] and [53] who reported a prevalence of 69.1% and 22%, respectively. One of the most likely hypotheses to explain the discrepancies is the differences in protocols used for the detection of thermophilic *Campylobacter*, and especially the absence of an enrichment step for the isolation of thermophilic *Campylobacter* in [54] work. In general, these variations might be due to approaches of sample collection, the difference in isolation and identification techniques, and differences in sample size.

The prevalence of *Campylobacter spp*. in goat meat was found to be 9.25%. This finding was in agreement with the reports of [9] and [55], who reported 7.6% and 6.4%, respectively. But it

**Table 5. In vitro antimicrobial sensitivity pattern of *C. jejuni* and *C. coli* isolates.**

| Antibiotic Discs | Interpretations | | | | | |
|---|---|---|---|---|---|---|
| | *C. jejuni* (N = 52) | | | *C. coli* (N = 12) | | |
| | S (%) | I (%) | R (%) | S (%) | I (%) | R (%) |
| Ampicillin | - | 2(3.8) | 50(96.2) | - | - | 12(100) |
| Amoxicillin | 3(5.8) | 4(7.7) | 45(86.5) | - | 2(16.7) | 10(83.3) |
| Chloramphenicol | 40(76.9) | 4(7.7) | 8(15.4) | 8(66.7) | 1(8.3) | 3(25) |
| Erythromycin | 43(82.7) | - | 9(17.3) | 9(75) | 1(8.3) | 2(16.7) |
| Gentamycin | 41(78.8) | 6(11.5) | 5(9.6) | 7(58.3) | 2(16.7) | 3(25) |
| Norfloxacin | 51(98.1) | - | 1(1.9) | 11(91.7) | 1(8.3) | - |
| Streptomycin | 22(42.5) | 7(13.5) | 23(44.2) | 3(33.3) | - | 9(66.7) |
| Tetracycline | 38(73.1) | 6(11.5) | 8(15.4) | 4(41.7) | - | 8(58.3) |
| Sulfamethoxazole- Trimethoprim | 39(75.5) | 3(5.8) | 10(19.2) | 7(58.3) | 1(8.3) | 4(33.3) |

S = Susceptible, I = Intermediate, R = Resistant.

slightly higher than the report of [42] who reported 4.4%. However, the present study was lower than the findings of [5] and [26] who reported 41.2% and 27.5%, respectively.

The meat samples collected from restaurants had the highest prevalence (43.93%). The meat samples collected from restaurants were found to be four times more likely to have *Campylobacter* compared to meat collected from Butcher and eight times more likely to have *Campylobacter* compared to meat collected from the abattoir. The difference in the prevalence of *Campylobacter* between sources of meat samples was found to be statistically significant (P<0.05) (OR = 7.64, CI = 4.01–14.52). This might be due to an extra chance of acquiring contamination from individuals who are working in restaurants during handling or cross-contamination among different carcasses.

In the current study, the bacteriological and PCR characterization of *Campylobacter* isolates revealed that the prevalence of *C.jejuni* was higher than *C.coli*. *C. jejuni* has been reported to be the most frequent species recovered from the food of animal origin especially chicken meat [48, 56–58]. The prevalence of *C. jejuni* and *C. coli* in bovine, goat, and chicken meat were found to be 76% and 24%; 80% and 20%; and 86.3% and 13.7%, respectively. These findings were in agreement with the findings of [9] who reported 78% *C.jejuni* and 18% *C.coli* [28]; who reported 78% *C. jejuni* and 22% *C.coli* [27]; who reported 93.3% *C. jejuni* and 6.7% *C. coli* [26]; who reported 72.5% *C.jejuni* and 27.5% *C. coli* in Ethiopia. The prevalence of *C.jejuni* in raw meat was in agreement with the reports from other countries [48, 57, 59, 60].

Antibiotic resistance in *Campylobacter* is emerging globally and has already been described by several authors and recognized by the WHO, as a problem of public health importance [61–63]. *Campylobacter spp*. resistance to antibiotics (*C.jejuni* and *C.coli*) can be transferred from different sources to humans. This situation, alarmingly, announces the need to perform an antimicrobial sensitivity test for *Campylobacter*. Macrolides and Fluoroquinolones are usually considered the drugs of choice for the treatment of foodborne Campylobacteriosis [64–66]. Antibiotic susceptibility patterns have been determined in previous studies conducted in Ethiopia where the 80%-100% of isolates from food animals were sensitive to these antimicrobial agents [9, 20]. However, there are pieces of evidence from different parts of the world that antimicrobial resistance in food animals and human isolates is increasing.

In the current study, 52 *C.jejuni* and 12 *C.coli* isolates were investigated for their antimicrobial susceptibility pattern. The percentage of ampicillin and amoxicillin resistant *Campylobacter* isolates were 96.9% and 85.9%, respectively. This was in agreement with the report of [28], who reported 97.2% and 83.3% for ampicillin and amoxicillin, respectively. Moreover, [67] reported a resistance level of 100% for *C. coli*. On the other hand, *C. coli* isolates are generally more resistant than *C. jejuni* strains [68]. In general, several studies have reported resistance to beta-lactam antibiotics is high in food animals [69–71]. The resistance rate of *Campylobacter* isolates (25%) to tetracycline in the present study was comparable with the findings of [27] (20.8%)but higher than that of [9] (10%) and [72] (6%). However, it was lower than the report of [73] (77.94%). The resistance level to streptomycin in the current study was 50%, which was higher than reports from Ireland and Thailand by [73] and [74].

Multi-drug resistance isolates always remained susceptible to norfloxacin and erythromycin and chloramphenicol. In the present study, multi-drug resistance to more than two antimicrobial agents was 68.7% which was comparable to the findings of [69] in Belgium, 60% [75]; in Estonia, 60% [49]; in Iran, 75%;and [76] in Korea, 93.4%. However, the current multidrug resistance finding was higher than the report from Addis Ababa and Debre Zeit, Ethiopia, by [9] (20%). Despite global commitments to reduce antimicrobial resistance and protect the effectiveness of antimicrobials, most countries have not yet started implementing government policies to reduce their overuse and misuse of antimicrobials [77]. Hence, the current multi-drug resistance finding might be since antibiotics can be bought for human or animal use

without a prescription, and similarly, in countries like Ethiopia without standard regulation and treatment guidelines, antibiotics are often over-prescribed by health workers and veterinarians and over-used by the public. Moreover, new resistance mechanisms are emerging and spreading globally. Hence, antibiotic resistance is rising to dangerously high levels in all parts of the world.

## Conclusion and recommendations

The present study revealed the occurrence of *Campylobacter* in bovine, goat, and chicken raw meat samples collected from different sites of the study area. Hence, they can serve as a potential vehicle for transmitting *Campylobacter spp.* and risk of infection to humans through the consumption of raw or undercooked meat. Therefore, retailers can act as a major source of cross-contamination. Moreover, the current study was revealed the development of antimicrobial resistance by the isolated *Campylobacter spp.* for certain drugs which is an alert for the concerned bodies. Hence, coordinated actions are needed to reduce or eliminate the risks posed by these pathogens at various stages in the food chain. Moreover, controlled and careful use of antibiotics, both in veterinary and human treatment regimes should be practiced. Finally, further nationwide molecular epidemiology and phenotypic and molecular characterization of the disease should be undertaken.

## Acknowledgments

The authors acknowledged owners, managers, and workers of the different abattoir, butcher shops, and restaurants of the study site for their keen interest and cooperation during the collection of the meat samples. We would like to extend our acknowledgement to the College of Veterinary Science Staff members' who were directly or indirectly helping us during the research period.

## Author Contributions

**Conceptualization:** Yohans Hagos, Getachew Gugsa, Nesibu Awol, Meselu Ahmed, Yisehak Tsegaye, Nigus Abebe, Abrha Bsrat.

**Data curation:** Yohans Hagos, Getachew Gugsa, Nesibu Awol, Meselu Ahmed, Yisehak Tsegaye, Nigus Abebe, Abrha Bsrat.

**Formal analysis:** Yohans Hagos, Getachew Gugsa, Nesibu Awol, Meselu Ahmed, Yisehak Tsegaye, Nigus Abebe, Abrha Bsrat.

**Funding acquisition:** Yohans Hagos, Getachew Gugsa, Nesibu Awol, Meselu Ahmed, Yisehak Tsegaye, Nigus Abebe, Abrha Bsrat.

**Investigation:** Yohans Hagos, Getachew Gugsa, Nesibu Awol, Meselu Ahmed, Yisehak Tsegaye, Nigus Abebe, Abrha Bsrat.

**Methodology:** Yohans Hagos, Getachew Gugsa, Nesibu Awol, Meselu Ahmed, Yisehak Tsegaye, Nigus Abebe, Abrha Bsrat.

**Project administration:** Yohans Hagos, Getachew Gugsa, Nesibu Awol, Meselu Ahmed, Yisehak Tsegaye, Nigus Abebe, Abrha Bsrat.

**Resources:** Yohans Hagos, Getachew Gugsa, Nesibu Awol, Meselu Ahmed, Yisehak Tsegaye, Nigus Abebe, Abrha Bsrat.

**Software:** Yohans Hagos, Getachew Gugsa, Nesibu Awol, Meselu Ahmed, Yisehak Tsegaye, Nigus Abebe, Abrha Bsrat.

**Supervision:** Yohans Hagos, Getachew Gugsa, Nesibu Awol, Meselu Ahmed, Yisehak Tsegaye, Nigus Abebe, Abrha Bsrat.

**Validation:** Yohans Hagos, Getachew Gugsa, Nesibu Awol, Meselu Ahmed, Yisehak Tsegaye, Nigus Abebe, Abrha Bsrat.

**Visualization:** Yohans Hagos, Getachew Gugsa, Nesibu Awol, Meselu Ahmed, Yisehak Tsegaye, Nigus Abebe, Abrha Bsrat.

**Writing – original draft:** Yohans Hagos, Getachew Gugsa, Nesibu Awol, Meselu Ahmed, Yisehak Tsegaye, Nigus Abebe, Abrha Bsrat.

**Writing – review & editing:** Yohans Hagos, Getachew Gugsa, Nesibu Awol, Meselu Ahmed, Yisehak Tsegaye, Nigus Abebe, Abrha Bsrat.

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
