## [Decision Letter · Decision Letter 0]

11 Nov 2020

PONE-D-20-31123

Isolation, Identification, and Antimicrobial Susceptibility Pattern of Campylobacter jejuni and Campylobacter coli from Cattle, Goat, and Chicken Meats in Mekelle, Ethiopia

PLOS ONE

Dear Dr. Gugsa,

Thank you for submitting your manuscript to PLOS ONE. After careful consideration, we feel that it has merit but does not fully meet PLOS ONE’s publication criteria as it currently stands. Therefore, we invite you to submit a revised version of the manuscript that addresses the points raised during the review process.

We look forward to receiving your revised manuscript.

Kind regards,

Kumar Venkitanarayanan, DVM, Ph.D.

Academic Editor

PLOS ONE

Journal Requirements:

2. We note that Figure 1 in your submission contain map images which may be copyrighted. All PLOS content is published under the Creative Commons Attribution License (CC BY 4.0), which means that the manuscript, images, and Supporting Information files will be freely available online, and any third party is permitted to access, download, copy, distribute, and use these materials in any way, even commercially, with proper attribution. For these reasons, we cannot publish previously copyrighted maps or satellite images created using proprietary data, such as Google software (Google Maps, Street View, and Earth). For more information, see our copyright guidelines: http://journals.plos.org/plosone/s/licenses-and-copyright.

2.1.    You may seek permission from the original copyright holder of Figure 2 to publish the content specifically under the CC BY 4.0 license. 

2.2.    If you are unable to obtain permission from the original copyright holder to publish these figures under the CC BY 4.0 license or if the copyright holder’s requirements are incompatible with the CC BY 4.0 license, please either i) remove the figure or ii) supply a replacement figure that complies with the CC BY 4.0 license. Please check copyright information on all replacement figures and update the figure caption with source information. If applicable, please specify in the figure caption text when a figure is similar but not identical to the original image and is therefore for illustrative purposes only.

3. We note you have included a table to which you do not refer in the text of your manuscript. Please ensure that you refer to Table 5 in your text; if accepted, production will need this reference to link the reader to the Table.

Reviewers' comments:

Reviewer's Responses to Questions

**Comments to the Author**

1. Is the manuscript technically sound, and do the data support the conclusions?

Reviewer #1: Yes

Reviewer #2: Partly

2. Has the statistical analysis been performed appropriately and rigorously? 

Reviewer #1: Yes

Reviewer #2: Yes

3. Have the authors made all data underlying the findings in their manuscript fully available?

Reviewer #1: Yes

Reviewer #2: Yes

4. Is the manuscript presented in an intelligible fashion and written in standard English?

Reviewer #1: Yes

Reviewer #2: No

5. Review Comments to the Author

Reviewer #1: Summary

In this manuscript by Hagos et. al., authors isolated Campylobacter spp. from cattle, goat and chicken meat sources in the city of Mekell of Ethiopia, and their antibiotic resistance profiling was performed. This study is significant since the Campylobacter infection is a major foodborne bacterial pathogen and their prevalence in various sources result in foodborne outbreaks. Also, these sources also can act as major hub for antibiotic resistance transfer. However, my major concern is that the selection of sample size for different meat type is not systematic or not defined well. And different meat types were having huge variation in sample sizes, provided, the study was conducted as a cross sectional study by actively collecting various meat types.

Major comments

1. Different meat types have different number of sample sizes, what was the basis of choosing the sample sizes for each category? Abstract says that the samples were collected from abattoir, butcher shops and restaurants, it should be clarified that how many samples are collected from each sourced for each meat type.

2. The ladder for PCR is not well defined and bands are not clear. Is there any reason why both C. jejuni and C. coli have different number of samples? Were they all collected from same meat type?

3. Authors discuss that the difference in prevalence of Campylobacter could be due to the difference in hygienic conditions in the chicken processing. Did the authors notice significant difference in hygienic conditions in processing various meats, if so, what were the differences?

Minor comments

1. The highest prevalence of Campylobcter is found in chicken meat, could it because of lower sample size of chicken meat? Was there any species difference between different meat types?

2. Some sentences are incomplete especially when quoting published articles, needs to double checked.

3. Reference formatting is not consistent. For example, page number in reference 52 compared to others.

Reviewer #2: Summary: The study investigated the prevalence of Campylobacter spp. in meat samples from various species followed by their antibiotic resistance profiling. The study is straightforward. Improvement is needed in the result and discussion sections. Please revise the manuscript based on my comments below.

Title: ok

Abstract: ok

Introduction: ok

Study area: ok

Study design: ok

Line 104-106: Sample size: Since it is known that chicken meat primarily harbors Campylobacter, why is the sample size of chicken meat less than cattle or goat. How was the sample size decided?

144: Typo- Mm or mM?

Results:

Question: How is the flow of meat till it reaches the restaurant? Is it Bucher to abattoir to restaurant? Why is butcher and abattoir a different category? Can the author comment on the significance of having butcher and abattoir separate? I am interested to see if during the supply chain, there is any change in the prevalence of Campylobacter. Interested to know what the authors think about this.

Figure 2: The quality of the figure is not publication standard. The 462 and 589 bp are too close to discern and the ladder is not well spread out. I recommend that the authors run the gel again. Maybe a thicker gel (2%) will help spread the bands out. It is also not clear if the c. coli or C. jejuni are randomly selected to run on the gel among the various strains available. Please explain.

Table 4 needs revision since some of the words are cut off from the manuscript. Also, please explain the numbers outside of the bracket. For example in Ampicillin resistant 2(96.9) what is 2?

Discussion:

Line 228-234: Can the authors comment on the sample size of studies where a higher Campylobacter prevalence was observed. How does the sample size affect the prevalence estimations?

Campylobacter in Cattle and goat meat: What is the potential source of Campylobacter in cattle and goat meat?

Line 258-263: Can the authors elaborate on why such a difference is observed in prevalence?

Line 264-272: The authors need to provide justification on the selection of the two genes to differentiate jejuni and coli. How are the two genes different? Where are they located on the genome? What is their function? Why are they separated based on species? What specific role do they play in the two species? Are they present on mobile genetic elements or DNA that can be exchanged between the species? These information needs to be added to the discussion.

Line 283-298: In its current form, the discussion on MDR or resistance in Campylobacter is very basic. Authors should include information on the timeline of antibiotic resistance analysis by the various studies and if there is any temporal changes observed. Of course, being from different geographical regions, a temporal correlation would be difficult but still an attempt should be made to observe trends. Also, the authors should comment on the potential route of exposure of Campylobacter to antibiotics in the various regions. IF there are regulations in place to reduce use of antibiotics and if that is bringing any change.

6. PLOS authors have the option to publish the peer review history of their article (what does this mean?). If published, this will include your full peer review and any attached files.

Reviewer #1: No

Reviewer #2: No

---

## [Author Response · Author response to Decision Letter 0]

26 Nov 2020

Manuscript PONE-D-20-31123

Response to Reviewers

Dear Publisher

Thank you for allowing us to submit a revised draft of the manuscript “Isolation, Identification, and Antimicrobial Susceptibility Pattern of Campylobacter jejuni and Campylobacter coli from Cattle, Goat, and Chicken Meats in Mekelle, Ethiopia” with an opportunity to address the reviewers’ comments for publication in your esteemed journal, PLOS ONE. 

Moreover, we would like to say You and both Reviewers many thanks for your invaluable time, critical and professional evaluation, as well as dedication to providing feedback on our manuscript, and are grateful for the insightful comments on and valuable improvements to our manuscript. We have a great appreciation for the corrections given by the reviewers and found them more valuable and professional for the betterment of our manuscript. 

Before we proceed to the response to the reviewers, let us explain the overall content of our manuscript and Journal requirement issues. 

1. Initially, we have prepared our manuscript according to the PLOS ONE style templates. 

2. We didn’t take Figure 1 (Administrative Map of Mekelle City) from other sources rather it is generated by us. Similarly, the image of Figure 2 (PCR Products: M-DNA ladder; Lane 1-4 C. coli (462bp); Lane 6-11 C. jejuni (589bp); and Lanes 5 and 12-Negative control) was taken by us.

3. In the revised manuscript, we have included/referred to Table 5 in the text.

We have incorporated most of the suggestions made by the reviewers though we have seen that the majority of comments require how and why explanations. Those changes are highlighted within the Revised Manuscript with Track Changes. Please see below, for a point-by-point response to the reviewers’ comments and concerns:

1. We have made some grammatical corrections as Reviewer 2 mentioned.

2. As Reviewer 1 stated the samples that were collected from the different meat types seem to be non-systematic and have a great variation among them. The reason for the variation in the number of samples collected from the different species of animals is that due to the accessibility/availability of the respected samples. Hence, the majority of bovine are slaughtered at abattoirs. However, the majority of goat and chicken are slaughtered at the home level without the recognition of the concerned bodies of the City and are most of the time inaccessible for researchers. Reviewer 1 also suggested us to indicate the numbers of samples collected from each sour, hence we have included under the sample size and sample collection section of the manuscript. 

3. Reviewer 1 mentioned that the ladder for PCR is not well defined and bands are not clear. It is quite right that the bands' resolution is not that much, however, we have mentioned under PCR section of the methodology as each band of the ladder has a molecular weight difference of 100bp. The samples were collected from different sources and we run the whole 64 phenotypically positive samples, hence, Figure 2 is a representative.

4. There were differences in hygienic conditions and evisceration of the internal organs which lead to cross-contamination.

5. Reviewer 1 mentioned why the highest prevalence is found in poultry? It is a well-known fact that poultry appeared to be a significant source of Campylobacter and chicken were found to be heavily intestinal carriers of Campylobacter when compared with other food animals in addition to what the reviewer suggested, sample size.

6. As per Reviewer 1’s comment, we have made corrections in the reference section.

7. Reviewer 2 needs an explanation for the small sample size of chicken meat which is also raised by Reviewer 1. So, we have indicated above in no. 2.

8. Reviewer 2 indicated that as there is an editorial problem in measurement, we have corrected it as per the given comment.

9. Reviewer 2 needs an explanation for the flow of meat till it reaches the restaurant. As per our Country’s practice, the animals are first transported to the abattoir to be slaughtered, and then the meat is either directly taken to the restaurant or the butcher shops depending upon the customer. As it is presented in Table 1, the highest prevalence was recorded in meat samples collected from restaurants. 

10. Reviewer 2 indicated that some words and numbers are cut off. So, we have corrected them. However, related to the number outside of the bracket it is not 2 rather it is 62, which means out of the 64 isolates 62 (96.9) had shown resistance against Ampicillin. 

11. Reviewer 2 needs an explanation of how sample size variation affects the prevalence estimation. In epidemiologic studies, the sample size has an important role to detect an effect and to achieve the desired precision in estimates of the parameter of interest. A small sample size will not provide a precise estimate and reliable answers to the study hypothesis. Hence, sample size variation may lead to a foregone conclusion in the prevalence of a disease.

12. Reviewer 2 has also asked the potential sources of Campylobacter in cattle and goat meat. It is known that Campylobacter species are normally carried in the intestinal tracts of many domestic livestock such as poultry, cattle, sheep, goat, pigs, as well as wild animals and birds. Fecal matter is a major source of contamination and could reach carcasses through a direct deposition. The meat of cattle and goat origin can become contaminated by this pathogen during slaughtering and carcass dressing. 

13. Reviewer 2 has requested us to elaborate on the possible reasons for variations in the prevalence of Campylobacter in the different sources of samples. It is an interesting comment. This might be because of an extra chance of acquiring contamination in restaurants due to mishandling of meat or cross-contamination among different carcasses or individuals who are working in restaurants might serve as a source of contamination.

14. Reviewer 2 requested us to explain about the two genes. Several multiplex PCR assays have been used to detect Campylobacter spp., C. coli, and C. jejuni. Most of these assays have used a variety of species-specific genes such as omp50, 16S rRNA, 23S rRNA, hipO, mapA, ceuE or putative aspartokinase for identification of C. coli and C. jejuni. The two commonly used species differentiation markers are mapA and ceuE . A species-specific 24-kDa membrane-associated protein of C. jejuni, named MAPA for membrane-associated protein is encoded by the mapA gene, which is useful for genetic identification of C. jejuni by PCR. ceuE encodes a 34.5 to 36.2 kDa lipoprotein component of a binding-protein-dependent transport system for the siderophore enterochelin. In the current study, for C. jejuni, the mapA gene was used and for C. coli, the gene ceuE was used.

15. Reviewer 2 has recommended us to include the timeline of antibiotic resistance as well as the potential route of exposure of Campylobacter to antibiotics in the various regions. Since we didn’t see the current study as per the given comment, we can’t do anything.

---

## [Editor Report · Decision Letter 1]

29 Dec 2020

PONE-D-20-31123R1

Isolation, Identification, and Antimicrobial Susceptibility Pattern of Campylobacter jejuni and Campylobacter coli from Cattle, Goat, and Chicken Meats in Mekelle, Ethiopia

PLOS ONE

Dear Dr. Gugsa,

Thank you for submitting your manuscript to PLOS ONE. After careful consideration, we feel that it has merit but does not fully meet PLOS ONE’s publication criteria as it currently stands. Therefore, we invite you to submit a revised version of the manuscript that addresses the points raised during the review process.

ACADEMIC EDITOR:

Please make the following revisions.

Line 69, Page

The situation seems to deteriorate more rapidly in developing countries where there is the widespread and uncontrolled use of antibiotics [22].

Delete “the” before widespread..

Line 79

goat, and chicken collected from the abattoir, butcher shops, and restaurants in Mekelle City.

Replace “the” with “an” if samples were collected from a single abattoir.

Line 87

The study was conducted from October 2015 to May 2016 at the abattoir, butcher shops, and

See previous comment.

Line 191-192

Contamination rate of C. jejuni and C. coli in the different samples  types (sub-heading)

Replace “samples types” with “sample types”.

Figure 1 PCR products

Separation of bands not clear. Replace the figure with a new figure showing band separation.

219-223

Species based antibiogram results of the isolates revealed that the high level of sensitivity of both  C. jejuni and C. coli was observed against Norfloxacin and Erythromycin, were 98.1% ………..as it is shown in Table 5.

Revise the statements for proper grammar.

270-273

This might be because of an extra chance of acquiring contamination in restaurants due to mishandling of meat or cross-contamination among different  carcasses or individuals who are working in restaurants might serve as a source of contamination

Revise the statements for proper grammar.

Reviewer 2 comment “Line 283-298: In its current form, the discussion on MDR or resistance in Campylobacter is very basic. Authors should include information on the timeline of antibiotic resistance analysis by the various studies and if there is any temporal changes observed. Of course, being from different geographical regions, a temporal correlation would be difficult but still an attempt should be made to observe trends. Also, the authors should comment on the potential route of exposure of Campylobacter to antibiotics in the various regions. IF there are regulations in place to reduce use of antibiotics and if that is bringing any change.

The authors have not responded adequately to the above comment and failed to revise the discussion. The authors are requested to respond/revise to the comments other than the timeline.

We look forward to receiving your revised manuscript.

Kind regards,

Kumar Venkitanarayanan, DVM, Ph.D.

Academic Editor

PLOS ONE

---

## [Author Response · Author response to Decision Letter 1]

5 Jan 2021

Manuscript PONE-D-20-31123

Response to Reviewers

Dear Publisher

Thank you for allowing us to submit the revised draft of the manuscript “Isolation, Identification, and Antimicrobial Susceptibility Pattern of Campylobacter jejuni and Campylobacter coli from Cattle, Goat, and Chicken Meats in Mekelle, Ethiopia” with an opportunity to address the reviewers’ comments for publication in your esteemed journal, PLOS ONE, for the second time. 

Moreover, we would like to say You and both Reviewers many thanks for your invaluable time, critical and professional evaluation, as well as dedication to providing feedback on our manuscript, and are grateful for the insightful comments on and valuable improvements to our manuscript. We have a great appreciation for the corrections given by the reviewers and found them more valuable and professional for the betterment of our manuscript. 

We have incorporated almost all of the comments given by the reviewers. Those changes are highlighted within the Revised Manuscript with Track Changes. Please see below, for a point-by-point response to the reviewers’ comments and concerns:

1. We have made the corrections given at Line 69, Line 79, Line 87, Line 191-192, Line 219-223, Line 270-273, and Line 283-298 as per the comments.

2. Regarding the comment related to the image in Figure 1, we have agreed with the reviewers’ idea. However, as we have already indicated in our previous response to reviewers, we can’t replace the figure with a new one since by the time the gel-doc was not working properly and we had taken the picture using our mobile phone’s camera. So, we have decided either to keep it as it is or to remove it from the manuscript. If you believe that as it is better to remove it, we don’t have a reservation on it. Hence, you can do that.

---

## [Editor Report · Decision Letter 2]

26 Jan 2021

Isolation, Identification, and Antimicrobial Susceptibility Pattern of Campylobacter jejuni and Campylobacter coli from Cattle, Goat, and Chicken Meats in Mekelle, Ethiopia

PONE-D-20-31123R2

Dear Dr. Gugsa,

We’re pleased to inform you that your manuscript has been judged scientifically suitable for publication and will be formally accepted for publication once it meets all outstanding technical requirements.

Kind regards,

Kumar Venkitanarayanan, DVM, Ph.D.

Academic Editor

PLOS ONE

---

## [Editor Report · Acceptance letter]

1 Feb 2021

PONE-D-20-31123R2 

Isolation, Identification, and Antimicrobial Susceptibility Pattern of *Campylobacter jejuni* and *Campylobacter coli* from Cattle, Goat, and Chicken Meats in Mekelle, Ethiopia 

Dear Dr. Gugsa:

I'm pleased to inform you that your manuscript has been deemed suitable for publication in PLOS ONE. Congratulations! Your manuscript is now with our production department. 

Kind regards, 

on behalf of

Dr. Kumar Venkitanarayanan 

Academic Editor

PLOS ONE